# Removal of Manganese(II) from Acid Mine Wastewater: A Review of the Challenges and Opportunities with Special Emphasis on Mn-Oxidizing Bacteria and Microalgae

**Yongchao Li [1], Zheng Xu [1], Hongqing Ma [2,\*] and Andrew S. Hursthouse [1,3,4]**

[1]  School of Civil Engineering, Hunan University of Science and Technology, Xiangtan 411201, China;
    nkliyongchao@163.com (Y.L.); jb1152929063@163.com (Z.X.); Andrew.Hursthouse@uws.ac.uk (A.S.H.)
[2]  Shandong Provincial Key Laboratory of Water and Soil Conservation and Environmental Protection,
    College of Resources and Environment, Linyi University, Linyi 276005, China
[3]  Hunan Provincial Key Laboratory of Shale Gas Resource Utilization, Xiangtan 411201, China
[4]  School of Computing, Engineering & Physical Sciences, University of the West of Scotland,
    Paisley PA12BE, UK
\*  Correspondence: hongqing6010@126.com

**Abstract:** Many global mining activities release large amounts of acidic mine drainage with high levels of manganese (Mn) having potentially detrimental effects on the environment. This review provides a comprehensive assessment of the main implications and challenges of Mn(II) removal from mine drainage. We first present the sources of contamination from mineral processing, as well as the adverse effects of Mn on mining ecosystems. Then the comparison of several techniques to remove Mn(II) from wastewater, as well as an assessment of the challenges associated with precipitation, adsorption, and oxidation/filtration are provided. We also critically analyze remediation options with special emphasis on Mn-oxidizing bacteria (MnOB) and microalgae. Recent literature demonstrates that MnOB can efficiently oxidize dissolved Mn(II) to Mn(III, IV) through enzymatic catalysis. Microalgae can also accelerate Mn(II) oxidation through indirect oxidation by increasing solution pH and dissolved oxygen production during its growth. Microbial oxidation and the removal of Mn(II) have been effective in treating artificial wastewater and groundwater under neutral conditions with adequate oxygen. Compared to physicochemical techniques, the bioremediation of manganese mine drainage without the addition of chemical reagents is relatively inexpensive. However, wastewater from manganese mines is acidic and has low-levels of dissolved oxygen, which inhibit the oxidizing ability of MnOB. We propose an alternative treatment for manganese mine drainage that focuses on the synergistic interactions of Mn in wastewater with co-immobilized MnOB/microalgae.

**Keywords:** Mn(II); acid wastewater; Mn-oxidizing bacteria; microalgae

## 1. Introduction

Manganese (Mn) is an essential nutrient for human life, and numerous enzymes utilize the redox properties of this element [1]. However, high levels of Mn in water supply stain porcelain and cause an undesirable taste in beverages [2,3]. Manganese is also toxic to humans when present in excessive concentrations in water. Emerging health studies are showing that its half-life in bones is about 8–9 years, once the Mn is absorbed by humans [4]. Intellectual impairment in school-age children, as well as hyperactive behaviors in school-age children exposed to Mn from drinking water, has been documented [5]. Even worse, Mn can be transported directly to the brain, leading to nerve

toxicity, resulting in anxiety, dementia, ataxia, and 'mask-like' face [6,7]. A disease called manganism is considered endemic in some South African manganese mines.

Given the adverse effects of Mn on human health, its release to natural freshwaters from anthropogenic sources is a major environmental problem. The World Health Organization in 2004 suggests a guidance value of 0.2 mg/L in freshwater. Many countries have also set limits for Mn concentration in water bodies. For example, the European Commission and the United States Environmental Protection Agency have set the Mn level in drinking water at 0.05 mg/L [8,9]. In China, the maximum level in surface water is 0.1 [10], and 5.0 mg/L in wastewater [11].

Manganese has been widely used in non-ferrous metallurgy, steel production, batteries, electrode materials, and catalyst production [12,13]. Most environmentally significant Mn contamination is directly related to the mining industry, where drainage waters have typically high concentrations of dissolved Mn [14,15] and Mn waste residues that can persistently release Mn into terrestrial and aquatic ecosystems [16]. South Africa, Russia, Australia, and China host the major Mn ore resources in the world [17]. It causes serious water and soil contamination, as well as human health risks [18,19].

Extensive research has been done on the development of effective methods to remove Mn from water. For acidic manganese mine drainage, increasing the solution pH and alkalinity are widely used techniques in Mn removal. Existing techniques for Mn removal from drinking or artificial water use sorption, oxidation, precipitation, and various physical, chemical, and biological processes. Manganese removal is challenging because of the complex chemistry of the element, and the characteristics of manganese mine drainage.

In this review, we first discus the Mn contamination sources from mineral processing and its potential impacts on the environment. Second, we discuss the available treatment options and detail the advantages and disadvantages of traditional techniques. Emphasis is placed on the biological oxidation of Mn(II) with Mn-oxidizing bacteria (MnOB). Finally, we propose a new remediation strategy for Mn removal.

*1.1. Sources of Contamination from Mineral Processing*

Manganese is released into the environment during the mining of Mn and other metals such as copper (Cu) and iron (Fe). In general, pyrite in the host rocks is the dominant problem because it creates acid conditions that solubilize Fe and Mn. Manganese oxide (pyrolusite) and Mn carbonate (rhodochrosite) minerals are the most important Mn ore resources in the world [17,20]. Gravity separation and flotation techniques are the most common processing methods for the separation of Mn ores [21]. After separation, recovery from low-purity ores for the production of superfine, hyperpure, and electrolytic materials [22] involves sulfuric acid extraction from Mn carbonate ore to produce Mn sulfate. This process is achieved via treatment by filtration, purification, evaporation, crystallization, and drying [23], or the reduction of Mn dioxide from pyrolusite [24].

Many deposits of low-grade Fe–Mn ore and rhodochrosite are simply abandoned. Generally, less than 60% of Mn ore is fully extracted. In Chinese production, the production of 1 tonne of pure Mn requires nearly 8 tonnes of pyrolusite/rhodochrosite ore. As a result, large quantities of tailings are deposited. With rainfall, potentially toxic elements associated with the ore deposits may be released from the tailings, so surface runoff is a major pollution source for local rivers and groundwater [25,26].

A large amount of wastewater containing Mn is also produced during the creation of electrolytic Mn for batteries. Manganese carbonate and Mn dioxide are raw materials used in electrolytic manganese production. High purity Mn is produced through leaching–purification–electrowinning. Wastewater from processing contains high concentrations of Mn(II), Cr(VI), and $NH_3$–N [27]. China plays an important role in global electrolytic manganese metal production, where its annual production capacity accounted for 99% of total world production in 2008. It appears that to produce 1 tonne of electrolytic manganese metal, approximately 1–3 tonnes of wastewater is discharged into the environment [28].

Generally, mine drainage from manganese mines is acidic. There are two main reasons. First, manganese carbonates are normally found in black shale containing large amounts of pyrite [29].

Secondly, due to the use of large quantities of sulfuric acid in the leaching process, the pH of wastewater is low [23,24]. Furthermore, the oxidation of Mn(II) may produce H ions to reduce the pH further [2]. Table 1 shows that the pH of manganese mine drainage ranges from 3.5 to 6.5, and the Mn ion content is high, especially in wastewater from electrolytic processing. In addition to Mn, high concentrations of other potentially harmful elements (mainly zinc (Zn), Cu, cadmium (Cd), Fe) are also found in the wastewater from Mn ore processing sites [15,30,31].

**Table 1.** Elemental composition of wastewater from Mn ore production.

| Location | Type of Wastewater | pH | Mn (mg/L) | Zn (mg/L) | Fe (mg/L) | Cu (mg/L) | Reference |
|---|---|---|---|---|---|---|---|
| Brazil | Mine water | 6.5 | 140 | 0.78 | 2.2 | 0.22 | [14] |
| Queensland, Australia | Mine open pit | 3.78 | 167 | 52.62 | 8.29 | 79.47 | [15] |
| Xiangtan, China | Runoff from tailings | – | 94.27 | 82.74 | – | 0.38 | [25] |
| Brazil | Mining wastewater | 3.54 | 99.42 | 16.82 | – | – | [30] |
| Chongqing, China | Electrolytic wastewater | 4.5 | 1805 | – | 2.1 | – | [32] |

The oxidation state of Mn can significantly influence its distribution, transport, and accumulation in water. Manganese can occur in several oxidation states: +2, +3, +4, +6, and +7. Mn(II) is the most common oxidation state in water with a pH lower than 7.0, while the more highly oxidized Mn(III, IV) exist at higher pH values and redox potentials [33,34]. Mn(VI) is not stable except in significantly basic solutions. Mn(VII), which usually exists as purple permanganate ion, is a powerful oxidizer and cannot be formed in most natural waters [34].

### 1.2. Influence of Manganese Mine Drainage on the Environment

Discharge of mine water without effective treatment results in long term effects on the environment. Examples include Zhunyi City, China, which hosts two manganese mines, and several smelting, grinding, and electrolysis plants, where some of the discharge drains directly into the Xiangjiang River. Manganese in sediments of the Changgou reach of the river was 100 times higher than the background value, where the pollution includes Cu, Fe, Zn, and Cd [35]. Huayuan County is known as "the eastern Mn city" in China, where the Huayuan River is the main source of drinking water for about 3 million people. However, the mean concentrations of Mn, Cd, and lead (Pb) all exceeded the acceptable levels for drinking [36]. It has been shown that soils in the Rongxi manganese mining area (Chongqing City, China) were extensively polluted, with total Mn, Cd, Pb, and Zn concentrations in the soils up to 48,400, 3.91, 80.68, and 131.23 mg/kg, respectively. Moreover, the contents of these metals in the dry shoots of plants ranged from 323–8434 mg/kg for Mn, 0.42–1.24 mg/kg for Cd, 3.90–105.84 mg/kg for Pb, and 19.17–57.64 mg/kg for Zn [16]. It was found that endangered wild northern quolls (*Dasyurus hallucatus*) living in manganese mining sites nearby, on the Groote Eylandt in Australia, had high-levels of Mn in their hair, testes, and two brain regions, which could affect the long-term viability of the population [37]. Owing to poor environmental awareness, lack of oversight of mine tailings, and poor Mn removal during treatment, mine wastewater could cause serious heavy metal pollution problems [38,39]. Therefore, the treatment of acidic manganese mine wastewater with high Mn(II) concentrations in the environment is important for living organisms and the future sustainability of several technologies.

## 2. Abiotic Treatment of Mn in Mining Wastewater

Abiotic treatments, including chemical precipitation, adsorption, ion exchange, chemical oxidation, and bioremediation technology, are often proposed to remove Mn from waters.

*2.1. Chemical Precipitation*

Chemical precipitation is a common method for metal ion removal from water bodies. Hydroxide, carbonate, and sulfide precipitation are the three main precipitation approaches used. Table 2 summarizes the separation of Mn(II) from contaminated waters.

**Table 2.** Summary of Mn(II) removal from wastewater using chemical precipitation methods.

| Type of Wastewater | Mn(II) (mg/L) | Precipitation Agent | Dosage (g/L) | Equilibrium pH | Temperature (°C) | Reaction Time | Removal Rate | Reference |
|---|---|---|---|---|---|---|---|---|
| Artificial solution | 2000 | NaOH | – | 9.0 | 60 | – | 99.5% | [40] |
| Mixed mine drainage | 1 289 | NaOH + $Na_2CO_3$ | 2.7 + 0.33 | 8.5–9.0 | Room temperature | 60 min | 99.6% | [41] |
| Acid mine drainage | 3.71 | Limestone | 2.94 | 10.0 | Room temperature | 6 h | 93% | [42] |
| Artificial solution | 155 | Limestone | 8.3 | 9.0 | 23 | 90 min | 99.4% | [43] |
| Mine water | 16.5 | Limestone | 16.7 | 9.0 | 23 | 90 min | 98.2% | [44] |

Hydroxide precipitation employs pH adjustment to convert heavy metal ions to less soluble hydroxide compounds. Given that solubility reduces as pH increases, Mn(II) can be removed by increasing the pH. Zhang et al. [40] used hydroxide precipitation to recycle Mn(II) from artificial wastewater. There was little manganese precipitation at pH < 6.0, but about 30% of the Mn was removed at pH 8.0 over 60 min. When the pH > 9, the Mn(II) concentration in the solution was reduced drastically from 2.0 g/L to less than 10 mg/L. Zn(II), Cu(II), and Mg(II) can also be removed using hydroxide precipitation in an aqueous solution.

Carbonate precipitation is also a practical process for dissolved Mn recovery. For example, at the wastewater treatment station of CITIC Dameng Mining Industries Limited (Daxi City, China), sodium carbonate was added to a tank containing 150 $m^3$ of manganese mine wastewater after the initial solution pH was adjusted to 8.5–9.0, where a frame filter removed the solid product. X-ray diffraction (XRD) analysis of the filter residue indicated its main component was $MnCO_3$, and some Mn(IV) was also found [41].

Limestone is the main precipitant because of its low-cost and availability [42]. For artificial solutions, 155 mg/L Mn(II) has been reduced to <1.0 mg/L with 8.3 g/L of limestone at an initial pH of 5.5, according to Equation (1), where K is the solubility product constant.

$$CaCO_3 + Mn^{2+} \rightarrow MnCO_3\downarrow + Ca^{2+} \qquad LogK = 2.35 \qquad (1)$$

However, the removal of 16.5 mg/L Mn(II) from acidic mining wastewaters was difficult and required an initial pH value of 8.0 and a larger limestone dose (16.7 g/L), probably due to the adverse effect of calcium (Ca) sulfate coating on the limestone [43].

In addition to the limestone, dolomite, magnesite, and quartzite can be used to precipitate Mn from mine wastewater. Soluble carbonate can be added if the solution alkalinity is not high enough for precipitation [40,43].

Metal sulfide precipitation is another important process in the hydrometallurgical treatment of ores and effluents. For example, sulfide precipitation with $(NH_4)_2S$ as a precipitation agent was successfully applied to the selective removal of Cu, cobalt (Co), and nickel (Ni) from Mn sulfate solutions [44]. However, metal sulfide precipitation is not commonly used for Mn(II) because the solubility of the MnS phases is quite high, which may result in the secondary release of Mn(II) from sediment to water. They are also about the toxicity of excess sulfide and unfavorable byproducts.

As shown in Table 2, precipitation can be applied very efficiently for high concentrations of Mn(II) once the solution pH > 9.0. However, the consumption of chemical reagents is high, and therefore, the operating costs increase. Additionally, large amounts of sludge can be produced and are commonly hard to dewater as a result of its amorphous particle structure [34].

### 2.2. Adsorption

Adsorption mechanisms are complex and may include adsorbate exclusion from the bulk solution, electrical attraction, and chemical complex formation on the adsorbent surface [45]. However, these operations are simple with minimal sludge production and regeneration capability.

#### 2.2.1. Adsorbents

Commonly used adsorbents for Mn removal include activated carbon [46], zeolites [47], kaolinite clay [48], nanoparticles [49,50], polymers [51,52], and a wide range of natural and artificial solids [53,54]. The search for low-cost adsorbents is a major focus of current research efforts, and a variety of natural minerals [15,55], agricultural and industrial wastes [56], and bio-sorbents [57] have been tested.

Table 3 assembles the maximum adsorption capacities ($Q_{max}$) for Mn(II) by adsorbents reported in the literature [15–58], which are predominantly from the Langmuir isotherm modeling of experimental data. Mn(II) adsorption mainly occurs in the pH range of 4.5~8.0, and the adsorption ability of kaolinite clay, pecan nutshell, nanoparticles, and Mn oxides was better than that of the other adsorbents. However, there was no similarity in the mechanisms of Mn(II) adsorption by these adsorbents. As for Nigerian kaolinite clay, the Elovich equation gave the best fit to the experimental data, and the presence of an intraparticle diffusion mechanism was indicated, although it was not the sole rate-determining step [48]. As for nanoparticles made from the mushroom *Pleurotus ostreatus*, the equilibrium sorption data were fit to the Langmuir model, indicating that a monolayer of Mn(II) ion was adsorbed on to the homogeneous adsorption sites of *Pleurotus ostreatus* nanoparticles. There, intraparticle diffusion and boundary layer diffusion were the rate-limiting steps [50]. However, the adsorption mechanism of Mn(II) ion by the pecan nutshell was found to be an interaction of active phenolic and carboxylic groups as the adsorbent of the metallic ion [56]. Three synthetic $MnO_2$ phases (pyrolusite, cryptomelane, and todorokite) exhibited different adsorption capacities, where todorokite was significantly more effective at removing Mn(II) than the other two phases, probably due to an ion-exchange mechanism involving tunneling, which intercalated Mg into todorokite [15]. Although the characteristics of adsorbents and their removal mechanisms play an important role in their removal ability to lower Mn(II), the comparison of $Q_{max}$ values should consider factors such as temperature, concentration range, and especially, the solution pH.

**Table 3.** Maximum adsorption capacities of adsorbents towards Mn(II) and reaction conditions reported in the literature.

| Adsorbent | pH | Temperature (°C) | Initial Mn(II) (mg/L) | $Q_{max}$ (mg/g) | Reference |
|---|---|---|---|---|---|
| Tunneled manganese oxides | 5.5–7.5 | 22 | 25 | 75–80 | [15] |
| Activated carbon | 4.5 | 23 | 20 | 1.90 | [46] |
| Beads of zeolite A | 6.2 | 25 | 100–600 | 30 | [47] |
| Kaolinite clay | 6.0 | 26.85 | 100–500 | 111.11 | [48] |
| $Fe_3O_4$ nanoparticles | 8.0 | 25 | 50–500 | 94.23 | [49] |
| *Pleurotus ostreatus* nanoparticles | 6.0 | 25 | 50–200 | 130.63 | [50] |
| Polyamidoxime chelating resin | 6.0 | 25 | 55–1099 | 82.85 | [51] |
| Metakaolin based geopolymer | 6.1 | 30 | 20–800 | 72.34 | [52] |
| Co/Mo layered double hydroxide | 5.0 | 24.85 | 40–145 | 20.20 | [53] |
| Functionalized polysilsesquioxane | 5.0 | 24.85 | 28–110 | 8.24 | [54] |
| Milled vermiculites | 6.8 | 25 | 5–300 | 33.73 | [55] |
| Pecan nutshell | 5.5 | 25 | 100–300 | 103.80 | [56] |
| *Pseudomonas putida* (wet biomass) | 6.5 | 24.85 | 4 | 0.22 | [57] |
| Polyvinyl alcohol/chitosan | 5.0 | 30 | 5–100 | 10.52 | [58] |

### 2.2.2. Effect of pH on Adsorption

Solution pH has a distinct effect on adsorption. Dawodu and Akpomie [48] investigated the effect of initial solution pH on the removal of 100 mg/L Mn(II) ions from a solution with 0.1 g unmodified kaolinite. The authors determined that 10–28% of Mn(II) was removed at pH values 2–4 but increased to 50% at pH 5. Liu et al. [49] showed that the Langmuir maximum adsorption capacity of Mn(II) on the $Fe_3O_4$ nanoparticles was 64.27 or 94.23 mg/g at a pH of 6.0 or 8.0, respectively. *Pleurotus ostreatus* was fabricated into nanoparticles (containing hydroxide, alkane, amide, and carboxyl (–COOH)) using high-energy nano-impact grinding, and then it was used as a new nano-bisorbent for the removal of Mn(II) [50]. It demonstrated that the $Mn^{2+}$ adsorption capacity of *Pleurotus ostreatus* nanoparticles increased from 41.5 mg/g to 53.0 mg/g as the pH increased from 3.0 to 4.0, but there was limited additional change as the pH increased to pH 7.0. The same phenomenon was observed for a metakaolin based geopolymer [52] and polyvinyl alcohol/chitosan [58]. It can be concluded that Mn(II) adsorption is weak at lower pH values for most adsorbents because the concentration of $H^+$ ions is higher at a lower pH. This result leads to competition for surface sites with the positively charged $Mn^{2+}$.

### 2.2.3. Effect of Mn(II) Concentration and Temperature on the Adsorption Capacity of Adsorbents

Initial metal cations concentration was often evaluated as an important factor in the wastewater treatment systems. It was found that the adsorption capacity of *Pleurotus ostreatus* nanoparticles increased sharply from 39.33 mg/g to 130.63 mg/g as the initial Mn(II) concentration increased from 50 mg/L to 200 mg/L [50]. The reason was that when the initial Mn(II) concentration was lower, the availability of active sites of adsorbent was relatively high, thus resulting in a further increase of its adsorption capacity. However, other researchers [49,51] showed that the amount of Mn(II) ion adsorbed onto adsorbents increased as the initial Mn(II) concentration increased before reaching a plateau. The most important reason was that at higher initial Mn(II) ion concentrations, more Mn(II) would be bound to adsorption sites on the adsorbent, and then reach adsorption saturation.

The effects of temperature on the Mn(II) adsorption capacities of the adsorbents were investigated from 5 to 50 °C. For example, the adsorption capacities of the metakaolin-based geopolymer were increased from 30.06 to 32.94 mg/g as the temperature rose from 10 to 50 °C. The adsorption process was endothermic and spontaneous. The increasing values of Gibbs free energy of adsorption ($\Delta G°$), with an increase in the temperature, demonstrated that Mn(II) adsorption was favorable at higher temperatures [52]. Similar results have been found in Mn(II) adsorption with Nigerian kaolinite clay [48], and raw and milled vermiculites [55]. However, a decrease in the adsorption capacity of *Pleurotus ostreatus* nanoparticles was observed when the temperature rose from 20 to 30 °C [50]. Thus, the effect of temperature on Mn adsorption ability from different adsorbents was complicated, whereas temperature had little influence on the adsorption equilibrium.

Most research on adsorbent removal of Mn(II) is from artificial wastewaters. Effective Mn(II) removal from real wastewater may be more difficult to apply due to competitive adsorption from other ions in solution, strong acidity, a minimum binding of Mn(II) ions at a lower pH value, and colonization by an Mn-reducing microorganism. The adsorption capability of adsorbents is rapidly saturated in several hours; thus, desorption with acid or oxidizing agents will be necessary to regenerate the adsorbents. At the same time, desorption is not always a simple process, and the volume of produced mine wastewater is high. Therefore, remediation using adsorbents alone can be expensive.

### 2.3. Oxidation/Filtration

An alternative treatment technique involves Mn(II) oxidation and precipitation as Mn dioxides ($MnO_2$) or as Mn oxyhydroxides (MnOOH), followed by clarification.

### 2.3.1. Oxidation for Precipitation

It is difficult to remove Mn(II) ions through direct oxidation using dissolved oxygen under neutral pH conditions. Only when pH > 8.5, can the oxidation of Mn(II) occur [59]. Although the abiotic oxidation of Mn(II) is not thermodynamically favorable in acidic environments, oxidative precipitation of Mn(II) with the addition of $SO_2$/air is rapid at pH > 5.0. This treatment removes 99.5% of Mn(II) from an artificial solution at pH 6.5 [40]. However, the consumption of $SO_2$ produces a large amount of sulfuric acid, which needs to be neutralized before discharge.

Chemical oxidation of Mn(II) can be accelerated by adding strong oxidizing chemicals, such as chlorine dioxide, ozone, or potassium permanganate [60]. However, there are several drawbacks to the use of these strong oxidants. Ozone and chlorine dioxide need relatively complex onsite generation, and chlorine dioxide can produce byproducts (chlorite) that require dosages at least twice that of the predicted stoichiometry as a result of interference from dissolved organic matter. Permanganate use may lead to the production of particulate Mn, and its overuse may result in the release of soluble Mn(VII). Ferrate Fe(VI) is thought to be an alternative to other strong oxidants in water treatment. It was also shown that Mn(II) oxidation by ferrate followed a stoichiometry of 2 mol Fe(VI) to 3 mol Mn(II), according to Equation (2).

$$2Fe(VI) + 3Mn(II) \rightarrow 2Fe(III) + 3Mn(IV) \tag{2}$$

The dissolved organic matter did not seriously affect the oxidation reaction rate, showing certain selectivity of Fe(VI) for Mn(II). However, the addition of strong oxidizing agents and ferrate Fe(VI) could increase treatment costs for metal-rich drainage water and also lead to an enhanced re-pollution potential [61].

### 2.3.2. Filtration

A low-cost coarse media for Mn(II) removal from water by filtration has been developed. The removal mechanisms of filters generally cover physical entrapment, gravity settling, impaction, straining, interception, adsorption, and flocculation. Piispanen and Sallanko [62] introduced an Mn oxide coated-sand/anthracite filter medium to remove Mn(II) from groundwater, and they found that its removal rate reached 91% with lower than 0.02 mg/L Mn(II) remaining in the treated water. Furthermore, the treatment of simulated mine water containing 1.97 mg/L Mn(II) with 5% $KMnO_4$ modified Mn sand exhibited better removal ability compared to ordinary Mn sand [63]. The surface of the modified Mn sand was covered with a dense membrane, which had the virtue of a large surface area, uniform texture, and physical and chemical stability.

Since the 1970s, drinking water treatment plants in the Netherlands, United Kingdom, and Malaysia adopted aeration followed by a sand filtration treatment process to remove Mn [64–66]. It was found that when the original water was passed through a filtration layer after aeration, Mn oxides ($MnO_2$ or $Mn_3O_4$) formed on the surface of the filter material, which further accelerated Mn(II) oxidation, and therefore enhanced its removal.

## 3. Biological Treatment of Mn(II)

Biological oxidation mechanisms for Mn(II) are different from chemically mediated processes. In the course of chemical oxidation/filtration for Mn removal, a variety of microorganisms have been detected on filter grains. Most of them are MnOB, which are associated with the oxidation and precipitation of Mn(II) [67]. Compared to conventional physico-chemical treatments, the biological treatment process has the advantages of no chemicals, higher efficiency, lower operation, and maintenance costs [68]. MnOB can oxidize Mn(II) ions with oxidation rates that are up to five orders of magnitude larger than that of chemical oxidation in the pH range of 6.5–8.5 [69,70].

Since the 1980s, biological oxidation and removal of Mn(II) have broadly been implemented at many laboratories and facilities. At the pilot scale, two trickling filters using different fractions of



silicic gravel as support media were constructed for Mn treatment of potable water [71]. Experimental results indicated that Mn oxidation was caused by both heterogeneous catalytic paths and biological oxidation. Bench-scale biological filters were used to remove Mn(II) from groundwater. In the absence of MnOB, Mn(II) could not be removed by air, whereas greater than 90% of Mn(II) was removed when the filtration columns were inoculated with a *Leptothrix* strain [72].

At full-scale water treatment plants, biological aerated filters (BAF) using supporting media for biofilm formation, are extensively used in China, Germany, France, and Japan for the treatment of Mn(II) [66,73]. Hasan et al. [74] also studied the simultaneous removal of 99 mg/L $NH_4^+$–N and 5.9 mg/L Mn(II) from polluted drinking water with a BAF system, and they found that with a hydraulic retention time of 24 h, the removal efficiencies for $NH_4^+$–N and $Mn^{2+}$ were of 99.3% and 99.1%, respectively.

In the literature, biological oxidation and removal capacities of Mn(II) from solution are reported extensively [73,75–89] (Table 4). Biological removal of Mn has mainly been applied to the remediation of groundwater and artificial wastewater systems. Mn(II) was almost completely removed when its concentration was low. Even when the Mn(II) concentration reached 50–100 mg/L, the removal rate was still above 95%. However, the percentage Mn(II) removal decreased with the increasing concentration to 274.7 mg/L, probably owing to toxicity, which suppressed the oxidation activity of the microorganisms.

**Table 4.** Biological oxidation and removal capacities of Mn(II) from the solution reported in the literature.

| Main MnOB or Fungus | Wastewater Types | pH | Temperature (°C) | Initial Mn(II) (mg/L) | Removal Ability | Reference |
|---|---|---|---|---|---|---|
| *Bacillus* sp. | Tap water | 6.6 | – | 0.35 | 95% | [73] |
| *Lysinibacillus* sp. | artificial Wastewater | 7.0 | 37 | 54.94 | 94.67% | [75] |
| *Brachybacterium* sp. | artificial Wastewater | 7.0 | 28 | 10.99 | >95% | [76] |
| Filter film | Groundwater | 8.0 | 22.2 | 0.99 | 99.5% | [77] |
| *Bacillus* sp. | Seawater | 7.5 | 24 | 1.65 | >83.3% | [78] |
| *Leptothrix, Pseudomonas* | Groundwater | 6.7–7.0 | 9 | 1.2 | >91.7% | [79] |
| *Methylosinus* | Artificial wastewater | 7.5 | 25 | 0–35 | 0.49 kg/m³/d | [80] |
| *Bacillus* | Elevated underwater surface water | 8.1 | 4 | 5.49 | 1.76 μg/g/d | [81] |
| *Bacillus pumilus* | | 6.3 | 22 | 0.1–0.2 | >98% | [82] |
| *Leptothrix, Pseudomonas, Hyphomicrobium* | Simulated Groundwater | 7.2 | 17 | 1–1.2 | 97.7% | [83] |
| *Crenothrix* | Groundwater | 7.0 | 7.8 | 0.9-1.3 | 95.90% | [84] |
| *Pseudomonas putida* | Artificial wastewater | 6.8 | 28 | 2.0 | >95% | [85] |
| *CueO* enzyme | Artificial wastewater | 8.0 | 37 | 274.7 | 35.7% | [86] |
| Mn-oxidizing fungus | Artificial wastewater | 7.0 | 25 | 60.43 | >99% | [87] |
| *Paraconiothyrium* sp. | Artificial wastewater | 6.6 | 25 | 380 | – | [88] |
| Acclimatized consortium | Artificial wastewater | 7.0 | 30 | 109.9 | 98.7% | [89] |

*3.1. MnOB Oxidation of Mn(II)*

Manganese oxidizing bacteria (MnOB) are mostly distributed in the genera *Pedomicrobium, Pseudomonas, Bacillus*, and *Leptothrix* [75]. The iron-bacterium *Leptothrix discophora* is the most widely identified Mn oxidizer downstream from coal mines [90]. Mechanisms for microbial oxidation Mn(II) are divided into either indirect or direct oxidation. Indirect oxidation can be due to a change of the surrounding environment as a result of the microbial growth and metabolism that can chemically oxidize Mn(II) to Mn(IV) [91]. Direct oxidation refers to the oxidation of Mn(II) facilitated by MnOB through a

specific Mn oxidase enzyme to accelerate the oxidation process or through proteins, polysaccharides, and other macromolecular substances in cell membranes that bind, concentrate, and adsorb Mn(II) [92].

### 3.1.1. Enzyme Driven Direct Oxidation

The majority of chemoheterotrophic microorganisms oxidize Mn(II) predominantly via an Mn oxidase enzyme. Until now, the enzymes that have been reported to be related to the microbial oxidation of Mn(II) are mainly multicopper oxidases (MCOs), Mn peroxidase, lactase, and lignin degradation enzymes [93–95]. Studies have focused on the important role of MCOs, which are a family of enzymes that employ Cu as cofactors in coupling the oxidation of substrate to the reduction of $O_2$ to $H_2O$ [94].

Multicopper oxidase type enzymes *CumA*, *MnxG*, *MofA*, and *McoA* in some model organisms were responsible for Mn(II) oxidation [96–98]. Geszvain et al. [99] found that after deleting either *MnxG* or *MofA*, the strains were still able to oxidize Mn(II), while oxidation was completely lost upon removal of both *MnxG* and *MofA* genes. The results suggested that *MnxG* and *MofA* independently encode MCO. Su et al. [100] also demonstrated that *CotA* oxidized Mn(II) intracellularly and the *CotA*-overexpressing strain possessed the capacity to remove Mn(II). Moreover, multicopper oxidase *CueO* in vitro or in vivo was found to accelerate the biological oxidation of Mn(II). A *CueO*-mediated catalysis system was established to produce biogenic Mn oxides (BMO), which were identified to be γ-$Mn_3O_4$, consisting of 28.0% Mn(IV), 18.4% Mn(III), and 53.6% Mn(II) [86].

Although several research studies have proven that MCO type enzymes were responsible for Mn(II) oxidation, the specific mechanisms remain poorly illustrated. MCOs are involved with pigment formation, siderophore oxidation, and biopolymerization, the loss of which can indirectly result in the non-Mn(II)-oxidizing phenotype [94]. However, studies found that some Bacillus strains that had Mn(II) oxidation activity, did not produce the MCO [101]. The result suggested that a single Mn oxidase was not the active mechanism in all MnOB strains.

### Effect of pH on Enzyme-Driven Oxidation

Natural Mn(II) oxidation without MnOB occurs very slowly when the solution pH ranges from 6.0 to 9.0 [102]. As shown in Table 4, MnOB can oxidize and remove Mn(II) from waters over a wide range of temperatures (4–37 °C) and initial pollutant concentrations (<l mg/L to >100 mg/L), although biological removal of Mn(II) has normally been applied in neutral water, at a pH of 6.3 to 8.1. The pH affects cell growth and the production of oxidation enzymes.

Adams and Ghiorse [103] found that solution pH can affect the Mn(II) oxidation ability of *Bacillus* species, and the related enzymes had maximum activity at a pH value of 7.3. Zheng [104] isolated *Bacillus* sp. T1151 from the soil in Shandong province, China, at an initial pH of 7.0, and where the $Mn^{2+}$ oxidation rate was highest. However, Mn oxide was not detected when the pH was higher than 9.0 or lower than 4.0.

The effects of pH on cell growth of *Pseudomonas putida* strains MnB1, and its Mn(II) oxidation action, were investigated by Zhou et al. [105]. As the pH increased from 5.6 to 7.5, the lag phase and reaction time required for BMO generation were shortened. When the pH was 7.5, 10 mg/L of amorphous β-$MnO_2$ was produced after 30 h of cultivation. However, when the pH decreased to 5.6, only 2 mg/L of Mn oxides were produced after 40 h of cultivation. These phenomena indicated that neutral or slightly alkaline conditions were favorable for the activity of Mn(II) oxidizing enzymes.

The pH-activity profile on *CueO* showed that it had a strong Mn(II) oxidase activity in the pH range of 7.6–8.2 [86]. Miyata et al. [70] found that the optimal pH for Mn(II) oxidation was 7.0 using a purified laccase-like fungal enzyme from cultures of *Acremonium* sp. strain KR21-2. Thus, the literature indicates that the optimal pH for most Mn(II)-oxidizing microorganisms or Mn oxidases is about 6.5 to 8.5 and that strong acid and alkaline environments can inhibit Mn(II) oxidative activity.

Additionally, some MnOB were found in acidic environments. Mayanna et al. [106] identified six bacterial strains (*Bacillus altitudinis*, *Bacillus safensis*, *Brevibacillus reuszeri*, *Arthrobacter*, *Frondihabitans*,

and *Sphingomonas*) as MnOB in soil samples from a former uranium ore in Ronneburg, Germany, which had an acidic soil pH (4.7–5.1) and Eh values varying from 640 to 660 mV. They discovered poorly crystalline birnessite in the Mn oxide layers.

Effect of Dissolved Oxygen (DO) on Enzyme-Driven Oxidation

The microbial Mn oxidation process is always aerobic. Katsoyiannis and Zoubouli [59] investigated microbial Mn removal from groundwater using a filtration unit mediated by *Leptothrix ochracea* and *Gallionella ferruginea*, both of which are MnOB. The DO first rose to 3.8 mg/L from an initial concentration of 0.9 mg/L following aeration, and then it reduced to 2.0 mg/L in the effluent as oxygen was consumed by the MnOB to oxidize soluble Mn(II). Pacini et al. [67] also indicated that MnOB required a completely aerobic environment (DO > 5 mg/L) to precipitate Mn(II) in water.

The effect of oxygen concentration in the culture medium on the $Mn^{2+}$-oxidizing activity of MnOB (*Pseudomonas fluorescens* GB-1) was studied by Okazaki et al. [107]. During the early and mid-logarithmic growth phases, the $O_2$ concentration in the cultures appeared to be almost zero as a result of the high metabolic activity of the bacteria. While no Mn(II)-oxidizing activity was seen when the $O_2$ concentration was below 14% saturation in the late logarithmic phase, the oxidation rate increased as the $O_2$ concentration rose from 15 to 26% saturation, followed by a decrease at higher oxygen concentrations. Mn(II) oxidation kinetics of *Bacillus* sp. SG-1 in soft water, artificial water, and natural seawater were measured, and it was found that an increase in the Mn(II) oxidation rate with a DO concentration from 2 to 270 μM followed the Michaelis–Menten model (Km(Michaelis–Menten constant) = 12–19 μM DO) [108].

Effect of Organic Carbon on Mn(II) Bio-Oxidation

From previous studies, MnOB are phylogenetically diverse, and they are found in a wide variety of environmental systems, such as polluted water bodies, rivers and seas, and soils in mining areas. To date, most of MnOB are heterotrophic aerobic bacteria that grow on organic substances [80]. However, MnOB have also been detected in drinking water treatment systems or wells with low concentrations of organic substrates, indicating that some MnOB were oligotrophic [66,109]. However, it was reported that MnOB enrichment in a downflow hanging sponge reactor failed at high substrate concentration, owing to competition with other fast-growing heterotrophs [110]. Accordingly, a rich substrate may not be favorable for MnOB cultivation in practical engineering, and a better approach may be the use of very low concentrations of dissolved organics.

Mn(II) removal from manganese mine wastewater by MnOB has not been extensively studied. In general, the mines have low nutrients and high contents of metal ions. The data suggest that supplementary organic compounds could be added to reach levels that might enhance biological Mn(II) oxidation.

3.1.2. Adsorption on the MnOB

Adsorption of Mn(II) can occur on the surface of BMO or on the bacterium itself. For example, Tang et al. [75] indicated that an isolate *Lysinibacillus* sp., strain MK-1 from stream silt, showed sturdy growth at 1 mM of Mn(II) in the pH range of 7.0–7.4. After seven days of culturing, the total Mn removal reached 94.67%, out of which Mn(II) oxidation accounted for 55.94%, and the Mn(II) adsorption by MK-1 accounted for 36.23%. Wang et al. [76] isolated a deep-sea MnOB, *Brachybacterium* sp. strain Mn32, showing high oxidizing abilities on Mn(II). They found that Mn(II) was first oxidized to Mn(III)-intermediates, and then the BMO generated around the cell surfaces further adsorbed more Mn(II) from the solution.

Biogeochemically active BMO is an efficient sorbent and can sequester greater quantities of dissolved Mn(II). Furthermore, BMO can adsorb other co-existing pollutant metal ions, such as Pb(II), Cd(II), and Zn(II) [111], and oxidize different substances (e.g., ciprofloxacin, As(III), and 17α-ethinyl estradiol) [112–114]. Extensive studies suggest that the structure and shape of BMO are not the same

under different reaction conditions. The majority produced are amorphous nano-particulates [115], with high specific surface area and low isoelectric point [88].

### 3.1.3. Indirect Oxidation

Indirect oxidation is another mechanism for biological Mn(II) oxidation. This occurs when microorganism growth increases the pH, oxidation-reduction potential, DO in the environment, or consumes $CO_2$ or acid [116].

Hullo et al. [117] demonstrated that although *B. subtilis* contained *CotA*, which was similar to laccases, this protein did not play any role in Mn(II) oxidation. The oxidation occurred due to the increasing pH promoted by *B. subtilis*. Leucoberbelin blue I assay, Mn(II) oxidation by cell-free filtrate, electron microscopy, and energy-dispersive X-ray spectroscopy results indicated that *Serratia* collected from a manganese mine in the Iron Quadrangle region promoted Mn(II) removal in an indirect mechanism through the formation of Mn oxide precipitates around the cells [118]. Further studies confirmed that the strain was able to increase the pH of the medium from an initial 7.5 to above 8.0, which favored the biological oxidation of Mn(II) [119].

Richardson et al. [120] reported that cyanobacteria could promote Mn oxidation via an indirect mechanism because the concentration of DO in the microenvironment was significantly enhanced. Moreover, stable oxygen isotope tracer studies of Mn oxide minerals demonstrated that as much as 50% of the oxygen in biologically generated Mn (IV) oxides might come from dissolved $O_2$ [121]. Tran et al. [114] also suggested that after MCOs enzymes catalyzed Mn(II), and the instantaneous product Mn(III) generated was further oxidized to Mn(IV) to form BMO incorporating oxygen. Therefore, the increases in environment DO as a result of microorganism growth benefits Mn(II) oxidation. In addition, enzymatic extracellular superoxide production by a marine bacterium (*Roseobacter* sp. AzwK-3b) was responsible for biological Mn(II) oxidation, leading to the precipitation of Mn oxides via an Mn(III) intermediate.

## 3.2. Fungal Oxidation of Mn(II)

### 3.2.1. The Action of Fungi on Mn(II) Treatment

Apart from bacteria, fungi also play an important role in Mn(II) oxidation in some passive biological field systems. Isolated fungi have displayed extremely high resistance to Mn(II) toxicity as compared to Mn(II)-oxidizing bacteria [122].

In the UK, a prototype bioreactor filled with Mn(IV)-coated small stones efficiently removed Mn(II) from contaminated mine water (pH 4–6) over one year [123]. More importantly, two species of fungi were isolated in the bioreactor, and they were assigned to Pleosporales (Ascomycetes), including a known Mn(II)-oxidizing fungus, isolate IRB 20-1. Burgos et al. [124] studied the microbiological characterization of samples collected from two field Mn(II)-removal systems, and it showed that fungi constituted 88% of the Mn(II)-oxidizing cultures, whereas 12% were bacteria. From biologically active limestone treatment beds that are often used to remediate coal mine drainage containing 150 mg/L Mn(II) in the eastern United States, Chaput et al. [125] discovered a greater quantity of Mn(II)-oxidizing fungi; and phylogenetic analysis revealed nine different species belonging to *Dothideomycetes* and *Sordariomycetes*.

### 3.2.2. Mn(II) Oxidation Mechanisms and Product Characteristics

The mechanisms and pathways of Mn(II) oxidation to Mn(III) by Basidiomycota and Ascomycota have also been studied. However, compared to MnOB, the mechanisms responsible for Mn(II) oxidation by fungi and the ultimate formation of Mn oxides in the environment are poorly understood.

The primary enzymes implicated in fungal Mn(II) oxidation are Mn peroxidases and laccases or laccase-like metalloproteins [126]. Recently, the production of extracellular superoxide ($O_2^-$) during cell reproduction was identified as the oxidant for Mn(II) to Mn(III) in a common ascomycete [127,128].

It was also found that poorly-crystalline hexagonal birnessite-like β-$MnO_2$ was the primary product of Mn(II) oxidation by four different species of Mn(II)-oxidizing Ascomycetes fungi (*Pyrenochaeta* sp., *Plectosphaerella cucumerina*, *Acremonium strictum*, and *Stagonospora* sp.) that were isolated from acid coal mine drainage treatment systems [129].

### 3.3. Microalgae Oxidation of Mn(II)

#### 3.3.1. Microalgae Resistance to Acid and Metals

Microalgae are unicellular or straightforward multicellular prokaryotic cyanobacteria and eukaryotic photosynthetic microorganisms that are primarily found in the taxa *Cryptophyta*, *Chlorophyta*, *Euglenophyta*, *Chlorarachniophyta*, *Rhodophyta*, and others, many of which can grow in harsh environments [130], such as acidic waters [131,132]. For example, Cu mine discharge into the Rio Tinto River (Iberian Pyrite Belt), creates extremely low pH water (1.0–2.5) that hosts phototrophic microalgae. *Chlorella*, *Dunaliella*, *Chlamydomonas*, and *Euglena* are major biomass in the river [133]. Varshney et al. [134] showed that *Galdieria sulphuraria*, *Cyanidium caldarium*, and *Dunaliella acidophila* grew well when the pH was lower than 3.0, while *Nannochloropsis salina*, *Nostoc commune*, and *Chlamydomonas applanate* grew in the pH range of 3.0–9.0. At Taihu Lake, in the Yangtze Delta plain, *Scenedesmus quadricauda* grew better with higher photosynthetic rates in the pH range of 3.5–10, owing to both carbon dioxide and bicarbonate being actively transported by *Scenedesmus quadricauda* [135]. Furthermore, acidophilic algae exhibited significantly higher accumulations of heat-shock proteins, which may be an adaptive mechanism to the strongly acidic conditions [136].

Extremophiles develop either a physiological or genetic adaptation to resist toxicity from high dissolved metal concentrations. Microalgae adapt to high metal concentrations through the induction of reactive oxygen species, and they release the enzymes peroxidase, catalase, and superoxide dismutase [137]. Therefore, microalgae have strong acid and heavy metal resistance, further increasing their applicability to remediation solutions.

#### 3.3.2. Mn(II) Uptake by Microalgae

Microalgae can adsorb Cu, arsenic (As), Mn, and Zn as a result of binding groups on cell surfaces [138]. Manganese is an essential nutrient for phytoplankton, and low concentrations limit algal growth [139]. Macroscopic algae were also documented to precipitate Mn oxides. In Pinal Creek downstream from Cu mining in Globe Arizona, distinctive holdfasts of the green alga *Ulothrix* sp. oxidized Mn and participated in a complex ecosystem whereby many organisms were building an Mn deposit in the streambed [140]. To examine the relationships between unicellular alga growth, Mn(II) concentration, and oxidation, Knauer et al. [141] exposed *Scenedesmus subspicatus* to a wide range of ion concentrations. A large fraction was present both intracellularly and as extracellular Mn(III/IV) oxides generated from Mn(II) oxidation, indicating that photosynthetic algae could affect the Mn cycle, through changing soluble Mn(II) to intracellularly bound Mn and then to solid-state Mn(III/IV) oxides. The microalgae *Siderocelis* sp. also performs Mn(II) oxidation [142], and biogenic Mn-Ca oxide forms on the cell walls of macroalga *Chara corallin* [143].

#### 3.3.3. Effects of Microalgae on Environmental pH, Alkalinity, and DO

Algae not only accumulate Mn(II) efficiently, they can accelerate the oxidation of Mn(II) to Mn(IV) through indirect oxidation. For example, when dense populations of cyanobacteria were present in Oneida Lake (New York), manganese oxidation occurred because the microenvironment pH was higher than 9.0 [119]. Recently, Wang et al. [144] demonstrated that after *Desmodesmus* sp. WR1 was cultivated in BG-11 medium containing Mn(II), and with an initial solution pH of 7.0 and DO concentration of 8.0 mg/L, the algae-generated BMO was found. The result was because algal growth probably provided favorable conditions for Mn(II) oxidation by increasing solution pH and DO concentration.

They found that a maximum of 5, 13, and 11 mg/L of BMO was formed at initial Mn(II) concentrations of 6, 30, and 50 mg/L, respectively.

Microalgae Production of DO

Oxygen production during photosynthetic activity is an essential trait of most microalgae [145]. Tu et al. [146] demonstrated that when *Scenedesmus obliquus* was inoculated in sanitary sewage from Shenzhen University Town, the DO concentration increased from an initial 7.3 mg/L to 10.3 mg/L on the third day, and remained above 9 mg/L during the next four days. Biodegradation of methane, methanol, and glucose by microalgae–bacteria consortium was assessed by Bahr et al. [147]. They found that oxygen was produced by the microalgae and was further utilized as an electron acceptor for the bacteria degrading organic carbon.

Aerobic conditions are often applied in wastewater treatment, especially for biological Mn oxidation. In addition to traditional air diffusion or surface aeration processes, oxygen can be introduced through microalgae photosynthesis, resulting in lower operating costs [148]. Moreover, microalgae can protect bacteria from harsh environments, providing habitat for the bacteria, and releasing an organic carbon source for heterotrophic bacteria [149].

Microalgae Increase pH and Alkalinity

The generation of alkalinity and increase of pH levels is known to be the result of algal inorganic carbon assimilation. During the microalgal photosynthetic activity, organics were produced for cell growth, and superfluous $OH^-$ ions remained as $CO_3^{2-}$ and $HCO_3^-$ were consumed in the process [150]. Steinmann et al. [151] found that pH values in Mörlbach Lagoon (southern Germany) could reach 11.3 in the summer resulting in the high growth rate of algae under the influence of light and continuous nutrient inflow.

Although increasing pH values resulting from excessive algal growth can cause damage to plankton and macrophyte roots in the lagoon [152], alkalinity generation is useful for removing metals from acid mine drainage. It was shown that alkalinity generation by *Spirulina* sp. growth precipitated major metals ions (Zn, Pb, Fe, and Cu) in acid mine drainage as metal hydroxides. The pH of effluent increased to 9.0 from an initial level of 1.8 [153]. In addition, Sheoran and Bhandari [154] treated mine water using a blue-green algae-microbial consortium in a bench-scale treatment test, and net alkalinity increased from −125 mg/L to 197 mg/L as $CaCO_3$ within 24 h of retention. Moreover, 23–29% sulfate, 95% Fe, 84–86% Zn, and 28–45% Mn were removed as sulfide or hydroxide precipitation. Thus, the generation of alkalinity by microalgae is an alternative treatment technique for removing metal ions by precipitation.

## 4. Combining MnOB and Microalgae

Although many studies have explored the biological oxidation of Mn in artificial wastewater or groundwater, and several Mn oxidizing microorganisms have been screened from acidic soils in mine areas, there are few studies on the treatment of manganese mine wastewater using MnOB. There are several reasons: (1) Mn ore wastewater is more acidic, and its Mn(II) concentration is much higher than that of groundwater or artificial wastewater, leading to high toxicity to MnOB; (2) oxidation of coexisting ferrous ion in Mn ore wastewater resulting from pyrite dissolution results in oxidizing by $O_2$ leading to insufficient DO concentration to support biological oxidation; (3) compared to acidic soil, nutrient sources in Mn ore wastewater are low. These conditions limit the biological oxidation of Mn(II) from manganese mine drainage. While the pH value of wastewater and DO concentration can be improved by adding alkali lime or aeration, these procedures are costly and create secondary pollution. Thus, providing more effective methods to solve the acidity and low DO content is needed.

There are few studies on Mn(II) oxidation by microalgae alone in mine wastewater. Park et al. [155] examined the mine precipitates and their associated microbiota from several abandoned mine sites in Korea. Except for MnOB, in the birnessite forming environments from abandoned coal mines,

*Leptolyngbya* was the major Cyanobacteria that oxidized Mn(II) oxidation through the release of molecular oxygen.

Although active microalgae can promote the indirect oxidation of Mn(II), BMO production is not very high relative to the initial Mn(II) concentration. The outcome is obvious when the initial Mn(II) concentration is high. Additionally, the Mn oxidation efficiency of microalga *Desmodesmus* sp. is not as high as that of MnOB strains. Under optimum pH and DO concentrations, 50 mg/L of Mn(II) was completely oxidized by MnOB (Table 3).

Currently no systematic analysis and research exists to address how microalgae alone can affect the Mn(II) oxidation and precipitation processes. Whether the existence of active microalgae increases the oxidative activity of MnOB is unclear. Gene-level studies have yet to be undertaken or published.

### 4.1. Mn(II) Removal from Mn Ore Wastewater by Co-Immobilized MnOB/Microalgae

Biofilters or BAF commonly are used for biological Mn(II) oxidation during the treatment of groundwater or artificial wastewater [73,77,83,84]. During this procedure, MnOB immobilized by filtering media are directly exposed to the Mn(II) ions. However, for the treatment of manganese mine wastewater, the MnOB thrive in harsh environmental conditions containing toxic concentrations of Mn(II), strong acidity, and competition with other indigenous microorganisms resulting in a low survival rate by MnOB and weak Mn(II) oxidation rates [78,89]. Microalgae can accelerate Mn(II) oxidation and precipitation of MnOB, but they tend to wash out of the treatment systems [156].

Cell immobilization techniques have been suggested as a strategy to enhance the tolerance of microorganisms to high concentrations of heavy metals and low pH, as well as decreasing bacterial washout. For example, metal-resistant immobilized sulfate-reducing bacteria beads were successfully used in studies on acid mine drainage treatment [157]. Recent efforts have been carried out using mixed bacterial co-immobilization [158]. Shen et al. [159] co-immobilized a bacterium (*P. putida*) using a microalga (*C. vulgaris*) in Ca-alginate beads. The co-immobilized treatment indicated better removal of nutrients from wastewaters than any single treatment, revealing that symbiotic interactions occurred between the two microorganisms.

By co-immobilizing MnOB with microalgae in the same microenvironment, competition might be reduced or avoided. Perhaps the reduction of wastewater toxicity might ensue. More importantly, due to photosynthesis, pH values and DO concentrations in the media would increase, thereby providing suitable conditions for MnOB growth. Additionally, extracellular polymeric substances produced by microalgae can enhance bacterial growth rates. Therefore, constructing a co-immobilized MnOB/microalgae system with high efficiency for Mn(II) oxidation and low biomass may be the key to success for treating manganese mine wastewater using cooperative biological methods.

### 4.2. Synergistic Treatment Design

DO concentration in the culture medium has a serious effect on biological Mn(II) oxidation activity. Some water supply plants that use biological oxidation for the removal of Mn(II) use an air compressor to diffuse the air upward, thereby supplying oxygen for the growth of biofilm-forming bacteria in the formation of precipitates [160]. However, maintenance of a high aeration rate could increase energy consumption and operating costs. Aeration methods need to be optimized for efficiency to be economical.

This review of biological Mn removal has suggested a testable method for the synergistic treatment of manganese mine wastewater using co-immobilized MnOB and microalgae. Figure 1 presents a potential mechanism for co-immobilized bacteria/microalgae beads. The Mn(II) removal processes, product characterization, and coordination mechanisms between MnOB and microalgae could be used to create the optimal composition and design of co-immobilized beads and optimization of a potentially useful treatment process.

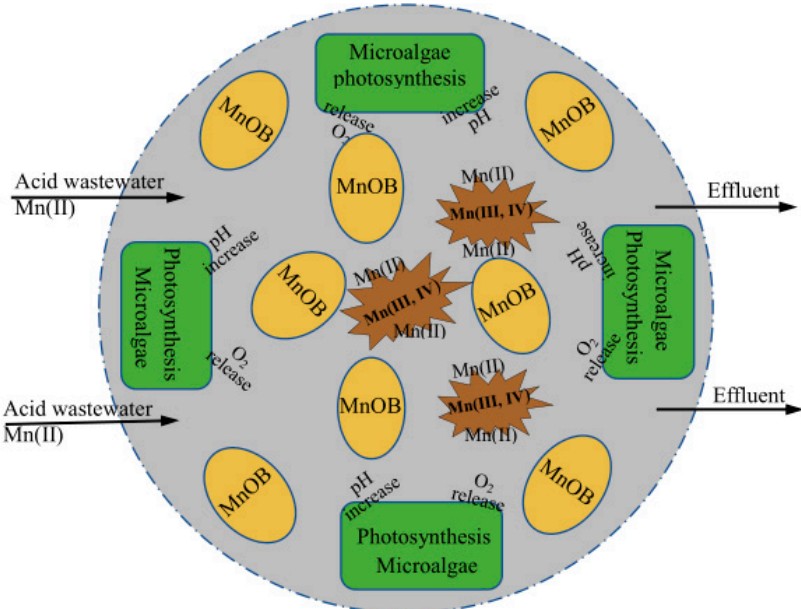

**Figure 1.** Speculative action and mechanisms for a manganese mine drainage treatment with co-immobilized bacteria/microalgae beads. MnOB denotes Mn-oxidizing bacteria.

The main steps in the development of the model include: (1) screening out oligotrophic MnOB from the Mn ore region; (2) clarifying Mn(II) oxidation mechanisms at a molecular level; (3) choosing coexisting microalgae that can thrive in acidic conditions; (4) preparation of co-immobilized MnOB/microalgae beads using environmentally friendly methods; (5) treating manganese mine drainage under various conditions using co-immobilized beads without external aeration, or neutralization of the wastewater.

Furthermore, combining the adsorption characteristics of BMO, coordination mechanisms for Mn removal by co-immobilized MnOB/microalgae beads is understood at the genetic level. Based on kinetic and thermodynamic studies of this removal process, it is possible to determine relationships between various environmental factors and Mn(II) oxidation rates, as well as the key steps in Mn removal.

## 5. Summary

The objectives of this review were to provide a comprehensive assessment of the main implications and challenges of Mn(II) removal from mine drainage. While chemical precipitation is the most straightforward method for Mn(II) removal from manganese mine drainage, the pH increase is expensive, and the resultant sludge may be toxic. Effective Mn(II) removal in actual wastewater is more challenging to apply because of competitive adsorption, strong acidity, and the regeneration of adsorbents. However, Mn(II) adsorption by low-cost adsorbents can be an alternative. Aeration/filtration identified as the primary mechanism responsible for the removal of lower concentrations of Mn(II) is often adopted by drinking water treatment plants.

This review also shows that removal of Mn(II) by MnOB is an effective method for wastewater treatment in neutral conditions, because Mn(II) can not only be oxidized to Mn(III,IV), but the resulting Mn(III,IV) oxide can further adsorb dissolved Mn(II). However, biogenic Mn oxidation is severely constrained at low pH. Moreover, aeration conditions are required during biological Mn(II) oxidation. As for Mn ore wastewater, its intense acidity and lack of sufficient dissolved oxygen mean that Mn(II) cannot be efficiently removed by traditional biological Mn oxidation.

Such limitations may be overcome through the use of microalgae, which are the leading primary producers in natural waters. Based on the review above, photosynthesis by microalgae can increase the phycosphere pH value and DO concentration, providing suitable conditions for Mn(II) oxidation in acid mine wastewaters. At the same time, microalgae could provide organic nutrient supplementation

for the growth of heterotrophic MnOB. More importantly, the important action of microalgae in Mn(II) oxidation and precipitation has been highlighted. However, the oxidation capacity of Mn(II) by microalgae alone was not very high.

It can be concluded that the introduction of selected microalgae into the traditional biological Mn oxidation system could help to solve the inhibition effect of high acidity and low DO on Mn(II) oxidation in acid wastewater. Thus far, no studies have reported on Mn ore wastewater treatment by the use of the MnOB/microalgae coexistence system. Meanwhile, the removal of Mn(II) using a MnOB/microalgae coexistence system, related coordination mechanisms, and process optimization problems remains to be tested.

## 6. Conclusions

Remediation of manganese mine drainage impacts is of particular concern to the scientific, business, and government communities. Practical measures to control manganese mine drainage have not particularly advanced. Indeed, Mn removal from wastewater generally relies on conventional physicochemical methods, such as precipitation, adsorption, and oxidation/filtration, using strong oxidation chemicals. However, these processes often exhibit operating problems and severe disadvantages, such as higher energy consumption and the generation of by-products. From the recent literature, biological oxidation and the removal of Mn(II) by MnOB has shown to be an effective method for the treatment of groundwater or artificial wastewater under neutral conditions. For Mn ore wastewater, its intense acidity and the lack of DO can inhibit Mn(II) removal using traditional biological Mn oxidation. However, these limitations may be addressed using microalgae because algal growth can increase solution DO levels and pH values, thereby offering optimum conditions for microbial Mn(II) oxidation in acid mine waters.

**Author Contributions:** Conceptualization, Y.L. and H.M.; methodology, Z.X.; validation, Y.L.; formal analysis, Z.X.; investigation, Z.X.; resources, H.M.; data curation, H.M.; writing—original draft preparation, Y.L.; writing—review and editing, A.S.H.; visualization, Y.L.; supervision, H.M.; project administration, Z.X.; funding acquisition, H.M.

**Funding:** This work was financially supported by the National Natural Science Foundation of China (No. 51504094), Natural Science Foundation of Hunan Province of China (2018JJ3177), and the Natural Science Foundation of Shandong Province (ZR2016EEB33).

**Conflicts of Interest:** The authors declare no conflicts of interest.

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
