# Peer review of "Removal of Manganese(II) from Acid Mine Wastewater: A Review of the Challenges and Opportunities with Special Emphasis on Mn-Oxidizing Bacteria and Microalgae"

_water, doi:10.3390/w11122493_

Round 1

Reviewer 1 Report

I edited this manuscript in a word file that won't attach here. 

Author Response

Dear Sir/Madam:

A thousand thanks for your comments and suggestions. I have already read the comments seriously and made substantial revisions to the manuscript according to these suggestions. Furthermore, a native English speaker who is familiar with the subject helped me to correct and rewrite the paper. Revised portions are marked in red in the paper.

Reviewer 2 Report

In this review of treatment methods for manganese mine wastewater, the
authors cover several types of treatment both abiotic
(precipitation, adsorption, oxidation/filtration) and biological
(bacteria, fungi, microalgae). The authors point out that techniques
combining bacteria and microalgae synergistically are particularly
promising, and worth exploring in future research.

The manuscript is well written and will be a useful addition to the
literature. I recommend publication after minor correction.

line 99: ``Generally, mine drainage from manganese mines is
acidic. There are two main reasons. First, manganese carbonate
deposits are normally found in black shale containing large amounts of
pyrite'' -- authors need to explain -- why does the presence of pyrite
confer acidity? is pyrite acidic?

line 160: equation (1) says LogK = 2.35; we should be told what K is
(a solubility product constant I would guess)

line 194: Pleurotus ostreatus nanoparticles -- my understanding that
these derive from an oyster mushroom; why aren't they mentioned as an
adsorbent in the list on line 190 along with kaolinite clay, pecan
nutshells, etc.

line 224: is it reasonable to report adsorption capacity in mg/g to 6
figure precision?

line 236: ``adsorption occurred favorable at higher temperatures'' --
this result surprises me -- in conventional adsorption studies, there
is less equilibrium adsorption at higher temperatures; typically
raising temperature shifts the equilibrium towards non-adsorbed states
which have higher entropy

line 396: I realise that the symbol Km is a Michaelis-Menten
constant, but the definition of the symbol should be mentioned

line 518 onwards: the point of sections 3.3.3.1 and 3.3.3.2 seems to
be to tell us that, over above their ability to accumulate Mn
directly, microalgae have side-benefits of creating an environment
conducive to removing Mn by other means; should we be told explicitly
that this is the point of these sections?

line 609: the authors propose a new synergistic treatment method for
manganese mine wastewater. However in the line immediately before
this, they mention that optimising the aeration method is an
issue. They don't mention how their new proposed synergistic treatment
method will address any issues with aeration method. Later on though
(line 622) they seem to suggest they might be able to treat manganese
mine wastewater without any external aeration at all. Can the authors
clarify?

Author Response

In this review of treatment methods for manganese mine wastewater, the authors cover several types of treatment both abiotic (precipitation, adsorption, oxidation/filtration) and biological (bacteria, fungi, microalgae). The authors point out that techniques combining bacteria and microalgae synergistically are particularly promising, and worth exploring in future research.

The manuscript is well written and will be a useful addition to the literature. I recommend publication after minor correction.

Response: Thanks. I have already read the comments seriously and made substantial revisions to the manuscript according to these suggestions.

line 99: ``Generally, mine drainage from manganese mines is acidic. There are two main reasons. First, manganese carbonate deposits are normally found in black shale containing large amounts of pyrite'' -- authors need to explain -- why does the presence of pyrite confer acidity? is pyrite acidic?

Response: Line 273-274, “In general, pyrite in the host rocks is the dominant problem because it creates acid conditions that solubilize Fe and Mn.” was added.

line 160: equation (1) says LogK = 2.35; we should be told what K is (a solubility product constant I would guess)

Response: line 950, “where K is solubility product constant” was added.

line 194: Pleurotus ostreatus nanoparticles -- my understanding that these derive from an oyster mushroom; why aren't they mentioned as an adsorbent in the list on line 190 along with kaolinite clay, pecan nutshells, etc.

"Mn(II) adsorption mainly occurs in the pH range of 4.5~8.0, and the adsorption ability of kaolinite clay, pecan nutshell, Fe3O4 nanoparticles and manganese oxides was better than that of the other adsorbents.” was changed into “Mn(II) adsorption mainly occurs in the pH range of 4.5~8.0, and the adsorption ability of kaolinite clay, pecan nutshell, nanoparticles and Mn oxides was better than that of the other adsorbents.”

“As for Pleurotus ostreatus nanoparticles, ……” was changed into “As for nanoparticles made from the mushroom Pleurotus ostreatus, ……”.

line 224: is it reasonable to report adsorption capacity in mg/g to 6 figure precision?

“It was found that the adsorption capacity of Pleurotus ostreatus nanoparticles increased sharply from 39.3337 mg/g to 130.625 mg/g as the initial Mn(II) concentration increased from 50 mg/L to 200 mg/L [49].” was changed into “It was found that the adsorption capacity of Pleurotus ostreatus nanoparticles increased sharply from 39.33 mg/g to 130.63 mg/g as the initial Mn(II) concentration increased from 50 mg/L to 200 mg/L [50].”

line 236: ``adsorption occurred favorable at higher temperatures'' --this result surprises me -- in conventional adsorption studies, there is less equilibrium adsorption at higher temperatures; typically raising temperature shifts the equilibrium towards non-adsorbed states which have higher entropy

Response: Thanks for your question. I have double checked the reference [48, 50, 52, 55]. It was true that within the certain scope of temperature, adsorption occurred favorable at higher temperatures

“The increasing values of Gibbs free energy of adsorption (ΔG°) with an increase in the temperature demonstrated that Mn(II) adsorption occurred favorable at higher temperatures [51]. Similar results have been found in Mn(II) adsorption with Nigerian kaolinite clay [47], raw and milled vermiculites [54], Pleurotus ostreatus nanoparticles [49]. So, a slight increase in the manganese adsorption was observed with increasing the temperature from 5 to 50 °C, whereas the temperature has little influence on the adsorption equilibrium.” was changed into “The increasing values of Gibbs free energy of adsorption (ΔG°) with an increase in the temperature demonstrated that Mn(II) adsorption was favorable at higher temperatures [52]. Similar results have been found in Mn(II) adsorption with Nigerian kaolinite clay [48], raw and milled vermiculites [55]. However, the decrease of adsorption capacity of Pleurotus ostreatus nanoparticles was observed when temperature arose from 20 to 30 °C [50]. So the effect of temperature on Mn adsorption ability by different adsorbents was complicated, whereas temperature has little influence on the adsorption equilibrium.”

line 396: I realize that the symbol Km is a Michaelis-Menten constant, but the definition of the symbol should be mentioned

Response: The definition of the symbol was added.

line 518 onwards: the point of sections 3.3.3.1 and 3.3.3.2 seems to be to tell us that, over above their ability to accumulate Mn directly, microalgae have side-benefits of creating an environment conducive to removing Mn by other means; should we be told explicitly that this is the point of these sections?

Response: In this review, microalgae not only have side-benefits of creating an environment conducive to removing Mn by other means, but also its photosynthetic activity can provide suitable condition for biological Mn(II) oxidation in acid manganese mine drainage with low DO content. This is reason why 3.3.3.1 Microalgae production of DO and 3.3.3.2 Microalgae increase pH and alkalinity was discussed.

line 609: the authors propose a new synergistic treatment method for manganese mine wastewater. However in the line immediately before this, they mention that optimising the aeration method is an issue. They don't mention how their new proposed synergistic treatment method will address any issues with aeration method. Later on though (line 622) they seem to suggest they might be able to treat manganese mine wastewater without any external aeration at all. Can the authors clarify?

Response: In 3.3.3.1. Microalgae production of DO, the oxygen production during photosynthetic activity of microalgae was discussed. It has been demonstrated that In addition to traditional air diffusion or surface aeration processes, oxygen can be introduced by microalgae photosynthesis, resulting in lower operating costs.

Reviewer 3 Report

The topic discussed in this review article is extremely important and needed since manganese removal from acidic effluence is still a serious environmental problem. The presented manuscript is written in a very transparent way and fully exhausts the topic. In the Reviewer's opinion, the article should be accepted in its present form.

Author Response

The topic discussed in this review article is extremely important and needed since manganese removal from acidic effluence is still a serious environmental problem. The presented manuscript is written in a very transparent way and fully exhausts the topic. In the Reviewer's opinion, the article should be accepted in its present form.

Response: Thanks for your recognition on our work.